# Folate and Vitamin B12 Status in Pediatric Hematopoietic Stem Cell Transplantation Patients

**DOI:** 10.3390/nu17030377

**Published:** 2025-01-21

**Authors:** Gizem Zengin Ersoy, Begüm Şirin Koç, Suar Çakı Kılıç

**Affiliations:** Pediatric Hematology & Oncology and Stem Cell Transplantation Department, Umraniye Training and Research Hospital, Health Sciences University, 34668 Istanbul, Türkiye; begumsirins@hotmail.com (B.Ş.K.); suarck75@gmail.com (S.Ç.K.)

**Keywords:** hematopoietic stem cell transplantation, vitamin B12, folic acid, pediatric transplantation

## Abstract

Background/Objectives: Vitamin B12 and folic acid (FA) are crucial for children’s hematopoiesis after hematopoietic stem cell transplantation (HSCT). This study evaluates the B12 and FA level changes before and after HSCT. Methods: We retrospectively collected data from 125 patients who underwent HSCT between March 2019 and February 2024. B12 and FA levels were measured at three time points: before transplantation, one month after, and six months after. Sixty-two patients had complete data. B12 deficiency was defined as levels < 200 pg/mL and insufficiency as 200–300 pg/mL. Folate deficiency was classified as insufficiency ≤ 3 ng/mL, indeterminate for 4.0–5.8 ng/mL and typical for 5.9–26.8 ng/mL. Patients with B12 < 300 pg/mL and folic acid < 5 ng/mL were treated at all stages. Results: Among the 62 patients, 24 (38.7%) were girls, with a median age of 4 years (1.75–8.25). Median B12 levels were 398 pg/mL (pre-transplant), 892 pg/mL (1 month post), and 430 pg/mL (6 months post). The second time point had significantly higher B12 levels (*p* < 0.001). Median folate levels were 9.7 ng/mL, 6.95 ng/mL, and 11.3 ng/mL at the respective time points (*p* < 0.05), with the second time point significantly lower. Conclusions: Pediatric HSCT patients experience increased demands for B12 and FA despite accurate treatment, leading to potential deficiencies. Close monitoring, early supplementation, and supplementing high levels of these micronutrients are essential.

## 1. Introduction

Hematopoietic stem cell transplantation (HSCT) is essential as a curative treatment option in treating many diseases, such as immunodeficiency, metabolic disease, bone marrow failure, and malignancies in childhood [1]. Since pre-transplant nutrition and nutritional vitamin levels affect post-transplant complications and outcomes, it is imperative to determine the patient’s nutrition status before HSCT. In addition, the conditioning regimen involving intensive chemotherapy and total-body irradiation (TBI) can cause serious side effects such as mucositis, nausea, vomiting, and diarrhea. It may result in inadequate oral intake, extensive malabsorption, and malnutrition through the gastrointestinal tract [2]. Close monitoring of the patient’s clinical status from a nutritional point of view plays a vital role in addressing HSCT conditioning regimen-related issues and reducing post-transplant mortality [3]. However, the continued growth and development of the pediatric patient group makes them more susceptible to nutritional deficiencies [4].

Vitamin B12 and folic acid (FA) deficiency in children can cause megaloblastic anemia, slowed growth and short stature, increased risk of infection, cognitive dysfunction, neurological damage, and, in severe cases, brain atrophy [5,6]. B12 and FA are tightly linked through their collaborative roles in one-carbon metabolism, and the hematologic complications seen in the deficiency of both vitamins are indistinguishable. Both manifestations are caused by impaired DNA synthesis, which causes the S phase of the cell cycle to lengthen, causing maturation to stop [7]. Both molecules are closely related to the patient’s pre-transplant nutritional status. They are of clinical importance for the patient as they are involved in the post-transplant cell cycle and hematopoiesis. Since transplantation is a process in which frequent infections and transplant-related complications can be seen, inflammation must not affect the micronutrients to be analyzed. According to a recent study, B12 and FA levels are not associated with inflammation, so these values should be examined in transplant patients [8].

Our study posits that pediatric patients will increase their micronutrient consumption during stem cell transplantation due to elevated cell turnover. We excluded micronutrients like iron and ferritin, which could be affected by transplantation-related inflammation. We did not include vitamins B1, B6, C, E, K, or trace elements such as zinc, copper, and selenium, as these were only assessed when clinically necessary. Vitamin D was also excluded from the study since it is not involved in cell turnover, although patients were regularly monitored and supplemented as needed. The study measured vitamin B12 and FA levels before the transplant, the first month after, and the sixth month after. According to current national protocols, we treated deficiencies and insufficiency in all three stages. We aimed to investigate how vitamin B12 and FA levels changed with appropriate treatment in pediatric HSCT patients. The secondary objective is to analyze the relationship between vitamin levels and early transplantation outcomes.

## 2. Materials and Methods

### 2.1. Patients

Between March 2019 and February 2024, the data of 125 patients who underwent HSCT at the Pediatric Stem Cell Transplantation Center of S.B. Ümraniye Training and Research Hospital were retrospectively reviewed. The inclusion criteria of the patients in the study were: at least six months after the stem cell transplantation, not experiencing any significant complications during the transplantation process (active infection), and not having an active gastrointestinal tract (GIS) graft-versus-host disease (GVHD) (grade 3–4). Patients who died within six months of HSCT were excluded from the study. Patients with complete B12 and folic acid levels at three time points: (1) before transplantation, (2) the first month after transplantation, and (3) the sixth month after transplantation were included in the study. Patients with missing data in any period were excluded from the study. B12 and FA values were present in 90 patients in the pre-transplant period, 76 patients in the first month after transplantation, and 74 patients in the sixth month after transplantation. A total of 62 patients with data for all three periods were included in the study. Serum folate and vitamin B12 concentrations were determined using electrochemiluminescence immunoassay (ECLIA, Roche COBAS).

The reference ranges for vitamin B12 and folic acid levels were based on laboratory reference values used in the hospital [9,10]. Vitamin B12 levels < 200 pg/mL were determined as deficient, and 200–300 pg/mL was determined as insufficient. Folate levels were classified as a deficiency for ≤3 ng/mL, indeterminate for 4.0–5.8 ng/mL, and typical for 5.9–26.8 ng/mL.

Patients with B12 < 300 pg/mL and folic acid <5 ng/mL were treated according to age and weight (Figure 1).

The study was conducted according to the ethical standards of the Declaration of Helsinki, and informed consent forms were obtained from patients and parents, ensuring the ethical conduct of our research. The Scientific Research Ethics Committee of Istanbul Health Sciences University Umraniye Training and Research Hospital approved the study, further validating its integrity.

Hematopoietic stem cell transplantation was performed from autologous, allogeneic (fully matched related or unrelated), and haploidentic donors. The conditioning regimens were selected according to the guidelines of the European Bone Marrow Transplantation (EBMT). The types of GVHD prophylaxis received by the patients after transplantation were arranged according to the type of transplant and donor (cyclosporine A (3 mg/kg/g) and methotrexate (10 mg/m^2^) in transplants from unrelated donors). The dose of anti-thymocyte globulin (ATG) was determined according to the type of transplant and donor before transplantation.

### 2.2. Statistical Methods

Data were analyzed using IBM SPSS, version 29.0 (IBM Inc., Armonk, NY, USA). The Kolmogorov–Smirnov test was applied to determine the distribution of the data. It was observed that B12 and FA levels in all periods did not match the normal distribution, so the Friedman test was applied to evaluate the difference between periods. If a statistically significant result was found, a post hoc test was applied to determine where the difference originated. Descriptive statistics that did not fit the normal distribution were given as median (interquartile range, IQR: the difference between the 75th and 25th percentile of the data). Categorical data were presented as numbers (n) and percentages (%). In the analysis, a *p* < 0.05 value was considered statistically significant.

## 3. Results

Of the 62 patients included in the study, 24 (38.7%) were girls. The median age of the patients was 4 years (1.75–8.25). Twenty-five patients were diagnosed with immunodeficiency, 11 with acute leukemia, 3 with Hodgkin’s lymphoma, 2 with Non-Hodgkin’s lymphoma, 2 with solid tumor, 12 with hemoglobinopathy, 2 with aplastic anemia, 3 with bone marrow failure syndromes (Fanconi Aplastic Anemia and Diamond Blackphan Anemia), and 2 with metabolic disease. Allogeneic (fully matched related or unrelated) HSCT was performed in 54 patients, haploidentical transplantation was performed in 3 patients, and autologous transplantation was performed in 5 patients (Table 1). The mean neutrophil engraftment time was 14.2 days (min: 9, max: 22); the mean platelet engraftment time was 21.7 days (min: 11, max: 91). Grade 3–4 mucositis was seen in 19 patients (30%). The mean duration of mucositis was 10.5 days (min: 0, max: 18).

### 3.1. Vitamin B12 Levels

The median B12 levels were 398.5 (IQR: 449, Q3:726-Q1:277) pg/mL before transplantation. After replacing low vitamin B12 levels according to age-appropriate protocols following each measurement, the levels were determined to be 892 (IQR: 893, Q3:1387–Q1:489) pg/mL and 430 (IQR: 395, Q3:727–Q1:332) pg/mL in the second and third periods, respectively (Table 1). In the post hoc tests, it was seen that the B12 level of the second time point was statistically significantly higher than the first and third groups (*p* < 0.001 and *p* < 0.001, respectively), and there was no significant difference between the first and third groups (Figure 2).

#### 3.1.1. Age, Sex, and B12 Levels

Before transplantation, B12 levels in boys (Median: 398.5, IQR: 222) were lower than in girls (Median: 444, IQR: 893.5). In the first month, boys’ B12 levels were 864 (IQR: 606), while girls’ B12 levels were measured at 1029.5 (IQR: 1022), and girls’ values were found to be higher. In the sixth month, boys’ B12 levels were slightly higher at 446 (IQR: 370) than girls’ (Median: 385.5, IQR: 318). No significant relationship was found between pre-transplant B12 levels and age (*p* = 0.51).

#### 3.1.2. Critical B12 Level

The number of patients with B12 levels below 200 pg/mL was 5 (8%), 0 (0%), and 3 (4.8%) in the first, second, and third periods, respectively. Those with less than 300 pg/mL were 16 (26%), 6 (10%), 12 (19%), and those below 400 pg/mL were 31 (50%), 10 (16.1%) and 26 (41.9%), respectively (Table 1).

### 3.2. Folate Levels

The median folate levels before transplantation were 9.7 (IQR: 7.8, Q3:13.3–Q1:5.5) ng/mL. Although low folate levels were corrected according to age-appropriate protocols after each measurement, the levels were determined as 6.95 (IQR: 7.9, Q3:12.2–Q1:4.3) ng/mL and 11.3 (IQR: 13.5, Q3:20–Q1:6.5) ng/mL (*p* < 0.05) in the first, second, and third periods, respectively (Table 1). In the post hoc tests, it was revealed that the folate level of the second time point was statistically significantly lower than the first and third groups (*p* < 0.05 and *p* < 0.05, respectively), and there was no significant difference between the first and third groups (Figure 3).

#### 3.2.1. Age, Sex, and Folate Levels

Pre-transplant folate levels were higher in boys (Median: 10.05, IQR: 6.7) than in girls (Median: 8.15, IQR: 9.45). However, at the first month, girls’ folate levels (Median: 8.65, IQR: 7) were found to be higher than boys (Median: 6.1, IQR: 8.9). At 6 months, folate levels were measured as 11.5 (IQR: 13.2) in boys and 10.95 (IQR: 11.85) in girls.

A significant negative correlation was found between pre-transplant folate levels and age (r = −0.345, *p* = 0.006), indicating that folate levels decrease as age increases.

#### 3.2.2. Critical Folate Levels

While the rate of those with <10 ng/mL folate level before transplantation was 50%, this rate increased to 64.52% in the first month and decreased to 46.77% in the sixth month. The rate of individuals with a <5 ng/mL level was 12.9% before transplantation, 37.1% in the first month, and 11.29% in the sixth month.

### 3.3. B12 and Folate Levels and HSCT Outcomes

Vitamin B12 and folate levels and long-term outcomes such as survival, GVHD, and graft failure could not be analyzed because of differences in diagnosis, transplant type, and conditioning regimens in the study group. However, their relationship with early outcomes, such as engraftment time and mucositis severity, was compared.

The comparison of pre-transplantation B12 and folic acid levels with neutrophil and thrombocyte engraftment times and the severity of mucositis clearly showed no significant differences. (*p* > 0.05) (Table 2 and Table 3).

## 4. Discussion

The increased growth rate in childhood leads to increased demand for micronutrients such as vitamin B12 and folic acid. These vitamins have an essential role in the cell cycle and erythropoiesis. HSCT aims to engraft bone marrow stem cells successfully and establish a healthy hematopoietic system by creating new stem cells. Children treated with conditioning regimens that include chemotherapy or radiotherapy experience many complications that affect nutrient intake, absorption, metabolism, transport, and excretion of nutrients [11]. Patients on a neutropenic diet have lower daily micronutrient intake than the reference values for a regular diet [12]. Causes such as mucositis, GVHD, and infections in children with HSCT also contribute to this condition by causing decreased absorption. In addition, increased cell turnover due to hematopoiesis and mucositis resumption during transplantation increases the need for micronutrients such as vitamin B12 and folic acid.

The present study retrospectively collected the data from a pediatric patient group to determine an appropriate approach in terms of the micronutrient supplementation needs of the patients by examining the change in the values. Therefore, patients’ vitamin B12 and folic acid values were evaluated at three time points: before, at the first, and at the sixth months after transplantation. When standard supplementation protocol was applied, vitamin B12 value increased, but folic acid value decreased significantly in the first month after transplantation.

Since no previous study has examined the change in vitamin B12 and folic acid values in pediatric HSCT patients, our findings were compared with similar studies conducted in special-needs groups. One is the study conducted by Onur et al. on patients in palliative care centers [13]. In this study, the micronutrient levels were measured at a single time point. Vitamin B12 levels were <200 pg/mL in 1.1% of patients and <300 pg/mL in 3.2%. Vitamin B12 deficiency was detected in neurological and neuromuscular disease groups, and there was no correlation between age and B12 levels. Folate levels were below < 3.9 ng/mL in 5.1% of the patients and <5.8 ng/mL in 16.1%. While there was no correlation between folate levels and gender, there was a correlation with disease group and age. Folic acid deficiency was most common in the congenital and genetic defects and neuromuscular diseases groups [13]. In our study, 8% of the patients had vitamin B12 levels < 200 pg/mL, while 16% had <300 pg/mL. The rate of patients with a pre-transplant FA level below 5 ng/mL was 12.9%, and those with a pre-transplant level below 3 ng/mL was 3.2%. This difference is due to the malignant group among transplant patients. Malabsorption and micronutrient uptake disorders can be seen in these patients due to chemotherapy.

In the study conducted by Revuelta et al., cancer patients were prospectively monitored in terms of micronutrient blood levels and needs. In this study, vitamin B12 levels were monitored at the time of diagnosis and intervals of 3 months. While there was no significant difference between the groups, it was found that vitamin B12 deficiency was more likely to occur in patients with high body mass index and obesity despite increased calorie intake [14]. Vitamin B12 levels did not change significantly during the 18 months that the study was ongoing. In our study, it was observed that both B12 and FA values varied over 6 months. This result is associated with increased cell turnover after HSCT. In the study by Song et al., the vitamin B12 levels of patients who achieved remission and those who did not were compared; it was seen that the vitamin B12 levels were significantly reduced in the remission group [15]. The authors also compared the patients’ treatment regimens and showed that the chemotherapy regimen affected the vitamin B12 and B6 levels, while the FA level was unaffected. The present study found that folic acid levels decreased significantly, and vitamin B12 levels increased significantly at the end of the first month, which is the period when the toxic effects of the conditioning regimen and transplant complications were most intense. Since replacement therapy was given to patients with pre-transplant vitamin B12 levels of <300 pg/mL, their B12 levels were found to be higher in the first month after transplantation. However, FA replacement is only in patients with <5 ng/mL, which could not prevent the decrease in FA levels in the post-HSCT period.

In a study involving 225 kidney transplant patients, the average vitamin B12 level of the patients was 362.57 pg/mL, and the vitamin B12 deficiency rate was 14% [16]. This value is similar to our study. The frequency of diarrhea and mycophenolate mofetil (MMF) use in patients with B12 deficiency was also higher than in patients with normal B12 levels. In this cohort, similar to Revuelte’s study, it was noted that female patients with a vitamin B12 deficiency had a higher body mass index. In both study groups, it was shown that excessive macronutrient intake did not meet sufficient micronutrient intake [14,16]. The Pontes et al. study also compared two groups of immunosuppressive therapies, MMF and azathioprine (AZA), in kidney transplant patients. While there was no significant difference between MMF and AZA, vitamin B12 deficiency was more common in patients using MMF compared to individuals with adequate B12 intake [16]. The study linked vitamin B12 deficiency mainly to inadequate intake, high adipose tissue, and MMF use. Our retrospective cross-sectional study could not analyze the cause of folic acid and vitamin B12 deficiency. Since the immunosuppressive drug preference was similar in the study group and only three patients used MMF, no additional analysis on immunosuppressives was added.

In a study conducted in the USA on daily folic acid intake in adult HSCT patients, there was no clear relationship between disease recurrence, acute GVHD, and survival outcomes [17]. In the study conducted by Nannya et al. in 15 adult HSCT patients, the pre-transplant folic acid level was <3.1 ng/mL in 50% of the participants, and this rate decreased to 21% on the fourteenth post-transplant day. The authors attributed this result, which was the opposite of the result in our study, to the use of parenteral nutrition [18]. A weak correlation was found between folic acid levels and transplant outcomes in a meta-analysis investigating the need for vitamins in adult patients during HSCT [19].

In our study, patients with pre-transplant B12 levels of <300 pg/mL were treated. Therefore, the B12 levels were higher in the first month after the transplant. However, although B12 levels were increased, 9.7% of the patients had B12 levels between 200–300 pg/mL and needed treatment in the first month after transplantation. Vitamin B12 levels decrease significantly in the sixth month after transplantation, which shows that the increased need for vitamin B12 continues between the first and sixth months after transplantation despite engraftment. For this reason, close monitoring of micronutrients and supportive treatment should continue during this period. In addition, keeping the threshold values higher (e.g., 400 pg/mL for B12) in patient groups with special needs, such as stem cell transplantation, will also be a solution.

When folic acid levels were analyzed, folic acid levels were significantly lower in the first month after transplantation. This condition may be related to methotrexate, which some patients take as a GVHD prophylaxis. In addition, 16.1% of patients needed replacement six months after transplantation. However, it also shows that the <5 pg/mL threshold value for folic acid replacement remains low to start replacement. Starting folic acid replacement in HSCT patients earlier and at higher threshold values will prevent folic acid deficiency in the first month after transplantation.

Although many studies examine nutrition in HSCT patients, no study examines the changes in micronutrient values such as vitamin B12 and folic acid, which are frequently deficient in childhood and essential both in the hematopoietic system and neurodevelopmentally. Our study is significant because it examines these deficiencies in the pediatric patient group undergoing HSCT. However, due to the retrospective character of the study, the pre-transplant and post-transplant nutritional status of the patients could not be fully included in the study. The relationship between nutrition and micronutrient intake could not be analyzed. In addition, disease subgroups, conditioning regimens, immunosuppressive drug use, and transplant type may affect the outcomes. Therefore, there is a need to repeat the analyses with more micronutrient varieties in larger cohorts. However, even in this study, folic acid and vitamin B12, two critical micronutrients for childhood, changed in the post-transplant period. This finding indicates that close follow-up and supportive treatment of patients during the first six months are necessary. Since this is the first study conducted on HSCT patients, it will serve as a stepping stone for future studies.

## 5. Conclusions

Vitamin B12 and folic acid are essential micronutrients in the cell cycle. In cases where the hematopoietic system is restructured and formed, such as HSCT, the need for vitamin B12 and folic acid increases in pediatric patients, and their deficiencies are expected. In high-stress situations like stem cell transplantation in children, continuous monitoring of vitamin B12 and folic acid levels is essential. New and more extensive follow-up studies are needed to reevaluate the thresholds for starting supportive therapy, as our study suggests that they should be set at a higher level. Protocols used in healthy children are inadequate in HSCT patients due to increased consumption and impaired absorption. Planning prospective studies on this subject is essential for determining the correct levels and establishing the proper protocols for starting replacement.

## Figures and Tables

**Figure 1 nutrients-17-00377-f001:**
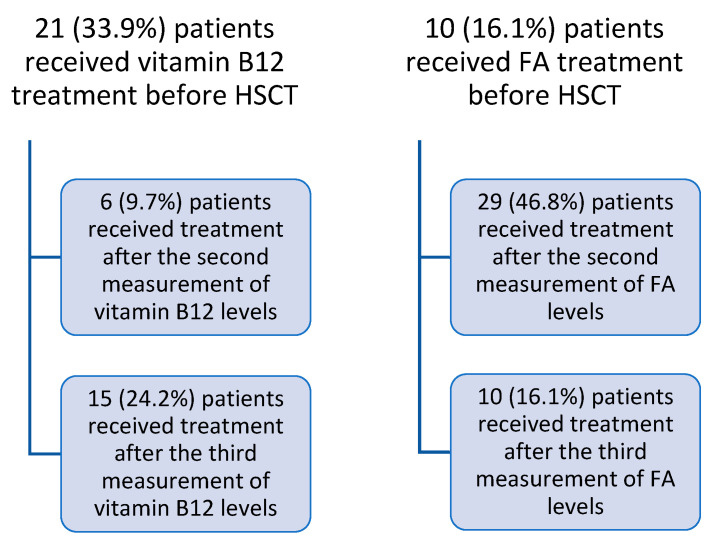
Treatment of vitamin B12 and FA insufficiency in all stages.

**Figure 2 nutrients-17-00377-f002:**
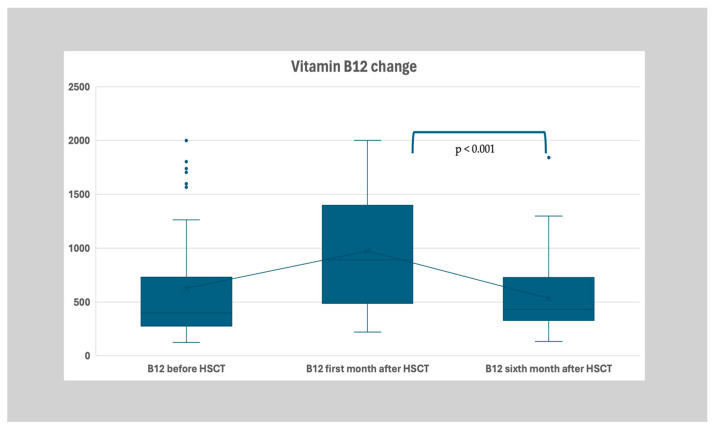
Distribution of vitamin B12 levels (pg/mL) before HSCT, the first month after HSCT, and the sixth month after transplantation. The B12 level of the second time point was statistically significantly higher than the first and third groups (*p* < 0.001 and *p* < 0.001, respectively).

**Figure 3 nutrients-17-00377-f003:**
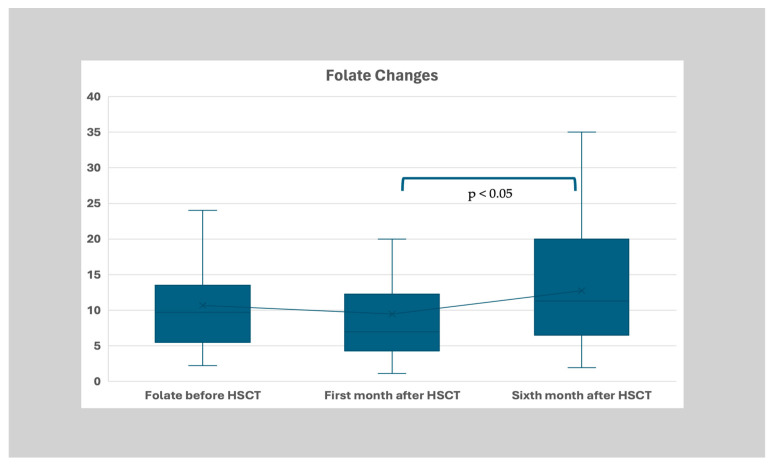
Distribution of folate levels (ng/mL) before HSCT, the first month after HSCT, and the sixth month after transplantation. The folate level of the second time point was statistically significantly lower than the first and third groups (*p* < 0.05 and *p* < 0.05, respectively).

**Table 1 nutrients-17-00377-t001:** Descriptive Statistics Table of Vitamin B12 and Folate Values of the Cases Before Transplantation, at first month and sixth months.

				Pre-HSCT Level	1st Month Level	6th Month Level
			n (%)	Median (IQR)	Median (IQR)	Median (IQR)
VITAMINE B12	Gender	Boys	38 (61.29)	398.5 (222)	864 (606)	446 (370)
		Girls	24 (38.71)	444 (893.5)	1029.5 (1022)	385.5 (318)
	Diagnosis	Immunodeficiency	25 (40.32)	488 (374)	978 (959)	413 (349)
		Hemoglobinopathy	12 (19.35)	326 (115)	1020.5 (774.5)	422.5 (563.5)
		Acute Leukemia	11 (17.74)	508 (920)	830 (1284)	430 (207)
		Hodgkin’s Lymphoma	3 (4.84)	593 (1457)	276 (195)	318 (651)
		Bone Marrow Failure	3 (4.84)	321 (196)	916 (1557)	359 (674)
		Aplastic Anemia	2 (3.23)	196 (146)	987 (958)	896.5 (803)
		Metabolic Disease	2 (3.23)	782.5 (723)	1114 (428)	743.5 (169)
		Non-Hodgkin Lymphoma	2 (3.23)	1019 (1368)	1011.5 (431)	599.5 (75)
		Solid Tumor	2 (3.23)	981 (1172)	847 (130)	479 (574)
	Transplantation Type	Allogeneic	54 (87.1)	393.5 (449)	989 (958)	436 (376)
		Haploidentical	3 (4.84)	386 (323)	675 (223)	396 (393)
		Autologous	5 (8.06)	593 (1024)	415 (506)	318 (545)
	All patients		62 (100)	398.5 (449)	892 (893)	430 (395)
	<400 pg/mL			31 (50)	10 (16.13)	26 (41.94)
	<300 pg/mL			16 (25.81)	6 (9.68)	12 (19.35)
	<200 pg/mL			5 (8.06)	0 (0)	3 (4.84)
FOLATE				**Pre-HSCT Level**	**1st Month Level**	**6th Month Level**
			**n (%)**	**Median (IQR)**	**Median (IQR)**	**Median (IQR)**
	Gender	Boys	38 (61.29)	10.05 (6.7)	6.1 (8.9)	11.5 (13.2)
		Girls	24 (38.71)	8.15 (9.45)	8.65 (7)	10.95 (11.85)
	Diagnosis	Immunodeficiency	25 (40.32)	11.7 (8.2)	10.2 (7.9)	17.3 (12.4)
		Hemoglobinopathy	12 (19.35)	7.3 (7.75)	10.45 (15.85)	6.35 (9.15)
		Acute Leukemia	11 (17.74)	7 (6.7)	4.5 (2.7)	10.6 (13.2)
		Hodgkin’s Lymphoma	3 (4.84)	6.1 (4.7)	4.5 (4.1)	6.9 (21.6)
		Bone Marrow Failure	3 (4.84)	9.4 (12.8)	3.7 (5.2)	13.4 (3.3)
		Aplastic Anemia	2 (3.23)	15.75 (8.5)	4.65 (5.7)	5.35 (3.7)
		Metabolic Disease	2 (3.23)	21.75 (20.5)	10.25 (1.5)	7.65 (10.7)
		Non-Hodgkin Lymphoma	2 (3.23)	3.05 (1.7)	26.65 (22.7)	6.7 (5.4)
		Solid Tumor	2 (3.23)	11.25 (0.70)	2.4 (2.2)	13.5 (13)
	Transplantation Type	Allogeneic	54 (87.1)	9.4 (9.1)	8.4 (8.6)	10.5 (12.8)
		Haploidentical	3 (4.84)	11.7 (4)	4.2 (0.9)	20 (0)
		Autologous	5 (8.06)	7.6 (4.8)	3.5 (3.2)	7 (13.1)
	All patients		62 (100)	9.7 (7.8)	6.95 (7.9)	11.3 (13.5)
	<10 ng/mL			31 (50)	40 (64.52)	29 (46.77)
	<5 ng/mL			8 (12.9)	23 (37.1)	7 (11.29)
	<3 ng/mL			2 (3.23)	6 (9.68)	3 (4.84)

HSCT: Hematopoietic Stem Cell Transplantation, IQR: Interquartile range (the difference between the 75th and 25th percentiles of the data).

**Table 2 nutrients-17-00377-t002:** The comparison of pre-transplantation B12 and folic acid levels with age of the patient, neutrophil and thrombocyte engraftment times.

		Pre-HSCT Vitamine B12	Pre-HSCT Folic Acid
Age	r	−0.085	−0.345
	*p*	0.51	0.006 *
Neutrophil Engraftment	r	0.026	−0.042
	*p*	0.841	0.743
Thrombocyte Engraftment	r	−0.226	0.035
	*p*	0.077	0.788

* *p* < 0.05 statistically significant value, HSCT: Hematopoietic Stem Cell Transplantation.

**Table 3 nutrients-17-00377-t003:** The comparison of pre-transplantation B12 and folic acid levels with severity of mucositis.

		Mucositis		
		Grade 1–2	Grade 3–4	*p* *
Pre-HSCT Vitamine B12	Mean ± Sd	662.4 ± 506.17	605.76 ± 543.65	0.366
	Median (Min-Max)	421 (152–2000)	395 (123–2000)	
Pre-HSCT Folic Acid	Mean ± Sd	9.8 ± 7.44	11.21 ± 5.25	0.084
	Median (Min-Max)	7.4 (2.2–32)	11.5 (3.8–24)	

* *p* < 0.05 statistically significant value, HSCT: Hematopoietic Stem Cell Transplantation, Sd: Standard Deviation, Min-Max: Minimum-Maximum.

## Data Availability

Data is available upon request from the corresponding author due to legal and ethical considerations.

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
