# Peer review of "Folate and Vitamin B12 Status in Pediatric Hematopoietic Stem Cell Transplantation Patients"

_nutrients, 2025, doi:10.3390/nu17030377_

Round 1

Reviewer 1 Report

Comments and Suggestions for Authors

Thank you for the opportunity to review the manuscript by Ersoy et al., which explores the role of folate and vitamin B12 status in pediatric bone marrow transplantation (BMT) patients. The topic is important, addressing nutritional and micronutrient deficiencies in this vulnerable population. Below are my comments for improving the manuscript.

The introduction provides a thorough overview of the nutritional challenges associated with BMT and highlights the relevance of micronutrient deficiencies. However, I believe that towards the end of the introduction, the authors should explicitly state their hypothesis and delineate their primary and secondary objectives. Currently, this section feels nonspecific and exploratory, lacking the precision needed to set the stage for the study.

The authors should alo explain why the study focuses solely on these two micronutrients and does not address others. Providing a rationale for this selection would enhance the manuscript's focus and relevance.

In the Materials and Methods section, the patient selection criteria are well-defined and clear. However, I would suggest the authors should include a reference for the values used to determine vitamin B12 and folate deficiency. Given the variability in standards, citing the source of these criteria would add rigor and credibility to the methodology.

The authors state that data are presented as median and interquartile range (IQR) in Table 1 and in Sections 3.1 and 3.2. However, only a single number is provided for the IQR. A range typically includes two values, representing the 25th and 75th percentiles. If the single number represents the difference between the 75th and 25th percentiles, this should be explicitly stated, as this is an atypical approach. I recommend presenting both the 25th and 75th percentiles, as this would provide a clearer and more conventional representation of the data.

The results are generally well-organized and presented. However, the issue with the IQR noted above is apparent in Sections 3.1 and 3.2. Clarifying this discrepancy or modifying the presentation to include both the 25th and 75th percentiles would strengthen the manuscript.

The discussion and conclusion sections are well-articulated, providing a thoughtful interpretation of the results. However, I have a concern about the authors recommending setting higher thresholds for initiating supportive therapy. Given the retrospective nature of the study and its small sample size, this recommendation seems overly strong. I suggest rephrasing this statement to indicate that such a "consideration" warrants further investigation in larger, prospective studies.

Finally, I have noted some grammatical errors and redundant sentence structures throughout the manuscript. Although the research is understandable even with these grammatical errors, I would suggest the authors refine the quality, or utilize language editing services to make improvements to an otherwise well-conducted and well-presented research study. 

Author Response

Comment: 1

The introduction provides a thorough overview of the nutritional challenges associated with BMT and highlights the relevance of micronutrient deficiencies. However, I believe that towards the end of the introduction, the authors should explicitly state their hypothesis and delineate their primary and secondary objectives. Currently, this section feels nonspecific and exploratory, lacking the precision needed to set the stage for the study.

The authors should alo explain why the study focuses solely on these two micronutrients and does not address others. Providing a rationale for this selection would enhance the manuscript's focus and relevance.

Response1:

We appreciate for this contribution we changed the last paragraph of the introduction section as follws: “Our study posits that pediatric patients will increase their micronutrient consumption during stem cell transplantation due to elevated cell turnover. We excluded micronutrients like iron and ferritin, which could be affected by transplantation-related inflammation. We did not include vitamins B1, B6, C, E, K, or trace elements such as zinc, copper, and selenium, as these were only assessed when clinically necessary. Vitamin D was also excluded from the study since it is not involved in cell turnover, although patients were regularly monitored and supplemented as needed. The study measured vitamin B12 and FA levels before the transplant, the first month after, and the sixth month after. According to current national protocols, we treated deficiencies and insufficiency in all three stages. We aimed to investigate how vitamin B12 and FA levels changed with appropriate treatment in pediatric HSCT patients. The secondary objective is to analyze the relationship between vitamin levels and early transplantation outcomes.”

Comment 2:

In the Materials and Methods section, the patient selection criteria are well-defined and clear. However, I would suggest the authors should include a reference for the values used to determine vitamin B12 and folate deficiency. Given the variability in standards, citing the source of these criteria would add rigor and credibility to the methodology.

Response 2:

Thanks for figuring it out we added references and stated it in the materals and methods section as follows: “The reference ranges for vitamin B12 and folic acid levels were based on laboratory reference values used in the hospital.[9,10]”

Comment 3:

The authors state that data are presented as median and interquartile range (IQR) in Table 1 and in Sections 3.1 and 3.2. However, only a single number is provided for the IQR. A range typically includes two values, representing the 25th and 75th percentiles. If the single number represents the difference between the 75th and 25th percentiles, this should be explicitly stated, as this is an atypical approach. I recommend presenting both the 25th and 75th percentiles, as this would provide a clearer and more conventional representation of the data.

The results are generally well-organized and presented. However, the issue with the IQR noted above is apparent in Sections 3.1 and 3.2. Clarifying this discrepancy or modifying the presentation to include both the 25th and 75th percentiles would strengthen the manuscript.

Response 3:

Thanks for your comment we clarifed the IQR clarified as “IQR: Interquartile range (the difference between the 75th and 25th percentile of the data)” under the Table1 and in the statistical methods section. The Q3 and Q1 results are added for each IQR in the sections 3.1 and 3.2.

Comment 4:

The discussion and conclusion sections are well-articulated, providing a thoughtful interpretation of the results. However, I have a concern about the authors recommending setting higher thresholds for initiating supportive therapy. Given the retrospective nature of the study and its small sample size, this recommendation seems overly strong. I suggest rephrasing this statement to indicate that such a "consideration" warrants further investigation in larger, prospective studies.

Response 4:

Thanks for the contribution we added a sentence to correct the statement as: “New and more extensive follow-up studies are needed to reevaluate the thresholds for starting supportive therapy, as our study suggests that they should be set at a higher level.”

Comment 4:

Finally, I have noted some grammatical errors and redundant sentence structures throughout the manuscript. Although the research is understandable even with these grammatical errors, I would suggest the authors refine the quality, or utilize language editing services to make improvements to an otherwise well-conducted and well-presented research study. 

Response 4: The text is rechecked and the grammatical errors are corrected.

Reviewer 2 Report

Comments and Suggestions for Authors

The manuscript is a valuable contribution to understanding micronutrient dynamics in pediatric HSCT. The findings emphasize the necessity of closer monitoring and tailored supplementation of B12 and folate in pediatric HSCT patients, providing actionable insights for clinicians However, its retrospective nature and limited focus on long-term outcomes constrain its impact. A follow-up study addressing these limitations would be a logical next step to validate and expand upon the findings.

Comments:
1. The study does not analyze long-term clinical outcomes (e.g., survival, GVHD severity, or neurodevelopmental implications), which limits the practical impact of the finding
2. The relatively small sample size and inclusion of a heterogeneous patient population (varied diagnoses and treatment regimens) make it difficult to generalize results.
3. The justification for the thresholds used to define deficiencies (<200 pg/mL for B12 and <5 ng/mL for folate) lacks discussion of why they might differ from those applied in healthy populations
4. While the study measures B12 and folate levels, it does not delve into potential causes of observed changes, such as absorption issues or drug interactions.
5. Missing data from several patients reduces the robustness of the analysis, and the absence of data on pre- and post-transplant nutritional interventions weakens the findings

Author Response

Comment1. The study does not analyze long-term clinical outcomes (e.g., survival, GVHD severity, or neurodevelopmental implications), which limits the practical impact of the finding.

Response1: Our study examined the long-term effects; however, we were unable to obtain significant results. This limitation was primarily due to the wide variety of root causes, including patient age groups, underlying diseases, conditioning regimens, and types of transplantation, all of which may influence long-term outcomes. Additionally, the low number of patients in certain subgroups further complicated the analysis. To maintain the article's fluency and integrity, we have chosen not to include these tests within the main text. If needed, we can provide this information as supplementary material.

Comment 2. The relatively small sample size and inclusion of a heterogeneous patient population (varied diagnoses and treatment regimens) make it difficult to generalize results.

Response2: Thank you for your valuable comment. We recognize that this is one of the main limitations of our article, which we have mentioned in the limitations section. However, our results still hold significant value as they will serve as a stepping stone for future studies.

Comment 3. The justification for the thresholds used to define deficiencies (<200 pg/mL for B12 and <5 ng/mL for folate) lacks discussion of why they might differ from those applied in healthy populations.

Response 3: We appreciate your valuable contribution. It is discussed between the lines 291-307 of the discussion section.

Comment 4. While the study measures B12 and folate levels, it does not delve into potential causes of observed changes, such as absorption issues or drug interactions.

Response 4. Thank you for identifying one of the limitations of our study. This limitation is also mentioned in the limitations section, specifically between lines 308-321. The retrospective design of our research presents challenges in analyzing nutrition and absorption issues. Additionally, the differences in underlying diseases, types of transplants, and conditioning regimens complicate the analysis of drug interactions. As a result, our study primarily focused on vitamin B12 and folic acid levels. In the future, we could continue the analysis with more homogeneous patient groups and delve deeper into the factors that may impair absorption.

Comment 5. Missing data from several patients reduces the robustness of the analysis, and the absence of data on pre- and post-transplant nutritional interventions weakens the findings

Response 5. We appreciate your valuable contribution. These analyses could not be conducted due to the retrospective design of our study. Our research can serve as a foundation for future studies, and nutritional effects can be incorporated in the next phase.
